# The Moral-Value Orientation—A Prerequisite for Sustainable Development of the Corporate Social Responsibility of a Security Organization

**Zdeněk Mikulka \***[ID] **, Ivana Nekvapilová and Jolana Fedorková**[ID]

Department of Leadership, University of Defence, Brno 662 10, Czech Republic;
ivana.nekvapilova@unob.cz (I.N.); jolana.fedorkova@unob.cz (J.F.)
**\*** Correspondence: zdenek.mikulka@unob.cz; Tel.: +420-724-605-294

**Abstract:** The article focuses on the social aspects of corporate social responsibility (CSR) in the Czech Armed Forces (CAF) and, more specifically, on professional ethics as a prerequisite for the sustainable development of the security organization. The text presents the results of research conducted on a sample of 278 members of the CAF. This research was based on Schwartz's holistic concept. To determine value orientation, a reduced version with 21 entries of the Schwartz's Portrait Values Questionnaire (PVQ) was used. Our data indicate that value orientation changes depending on military rank, depends, to a certain degree, on trait conformity (including obedience, respect for authorities, politeness, and self-control), and increases in the presence of lower-ranking individuals. Based on these findings the authors recommend to continue monitoring the value profiles of CAF members at various stages in their careers, to determine the optimal range of self-identification with a certain military rank and position, and to provide rank and position-specific educational programs into military ethics and ethical leadership aimed at sustainable development of moral-values.

**Keywords:** CSR; sustainable development; moral-value orientation; ethics; ethical responsibility

---

## 1. Introduction

The need to improve quality of life in civil society and to maintain the long-term development of a market economy has brought with it an emphasis on research into, and evaluation of, the conduct of large social actors such as enterprises. They represent important actors in social change and, like private individuals, are bound by responsibility towards the society within which they operate, and their actions are weighed against the moral compass of that society. Although the concept of corporate social responsibility (CSR) has been around for more than 40 years [1], there is no universally accepted definition yet [2]. This is, to a certain extent, because social responsibility has no clearly defined boundaries, either with regard to the business environment or to the environment of state and semi-state institutions and organizations. To date, it has largely been based on a voluntary commitment by these organizations [3,4].

In addition to companies, state organizations and especially security organizations such as the army, police, and components of the integrated rescue system, belong to the group of large social actors with a high impact on negotiations within the social unit. Although the management of such organizations has its peculiarities, the concept of CSR both offers them a number of opportunities for development and the opportunity to positively contribute to the sustainable development of society.

With regard to the Czech Armed Forces (CAF), the implementation of CSR principles paves the way for systemic change [4]. This change includes the reorientation from short-term goals to long-term visions and from a maximum (i.e., at all costs) emphasis task completion to a socially optimized

fulfilment of tasks, with regard for the needs of the internal and external environment. Accordingly, a socially responsible national security organization can contribute to sustainable development, be transparent, and help improve the quality of life in civil society within the limits of its social mandate [5].

The concept of CSR, as well as the concept of corporate sustainability (CS), have several implications [6]. For the state security organization, an important measure of CSR success is the degree of integration of socially desirable attitudes, practices, and quality programs that develop the organizational culture in conjunction with the organization's development strategy, at all levels of management [6–10]. This requires a change in perspective on the social role of the organization and a shift from a "profit only" level to a "people, planet, and profit" attitude [11,12]. This triple-bottom-line includes the attainment of the organization's goals, its development, and the fulfilment of its social obligations [12]. For the CAF this entails the realization that the organization is not insulated from the outside world and that, therefore, the success of its activities, at least in part, relies on how it is socially evaluated [13,14]. In this sense, the CSR CAF, in line with the European Union Green Paper, can be defined as "the voluntary integration of social and environmental considerations into day-to-day operations and interactions with corporate stakeholders" [15,16].

The application of the aforementioned CSR definition to the conditions of the CAF is based on the three pillars identified in the triple-bottom-line; performance (ensuring military and security activities), social, and environmental (see Dow Jones Corporate Sustainability Criteria contained in the EU Green Paper and OECD document) [16].

CSR in the field of military and security activities includes, but is not limited to, respect for the institutional code of ethics, transparency of processes, the application of the principles of corporate governance, the systemic rejection of corruption, the development of employee relations (shareholder dialogue), an open attitude towards the public, citizens, civil, and public institutions, and the protection of the intellectual property generated within the organization [17].

CSR in the social field includes the development of philanthropy [17,18] within the organization and its military and security activities, stakeholder dialogue [16,19], health and safety of employees during their professional activities, sustainable development of human capital, especially in the moral-value area, providing social security, adhering to labor standards, creating conditions for work-life balance [18,20], equal opportunities (for women and men and other disadvantaged groups in general), respect for ethnic diversity, ensuring retraining of redundant employees for their further employment, and respect for human rights [21,22].

Environmental CSR includes environmental protection of military and security activities according to International Organization for Standardization (ISO) 14,000 and Eco-Management and Audit Scheme (EMAS) standards, internal environmental policy (recycling, use of organic products), reduction of environmental impacts of military and security activities performed, and protection of natural resources [23,24].

Conducting oneself in accordance with the principles of CSR brings several advantages and benefits, especially in non-financial forms. The importance of these advantages and benefits, for the long-term functioning of the state security organization, cannot be overstated. Tangible assets in the form of real estate, inventory, and military material, as well as its financial security, are as important as intangible assets such as human capital, brand value, reputation, trust, and partnership [22,25].

Socially responsible organizations are characterized by a proactive policy of building and developing intangible assets that can bring greater social credibility, reputation, legitimacy, attractiveness, greater transparency, building socio-political capital, dialogue and building trust with civil society, increased loyalty and productivity of existing employees, the ability to attract and retain new quality employees, opportunities for innovation, the creation of facilities for smooth and successful operation and, last but not least, direct financial savings associated with effective military practice [24–27].

For the CAF to fully appreciate the benefits of CSR, its CSR must be genuine and credible. The following four axioms have been identified [28] as essential for CAF's credibility with the public:

- Personality—CAF must attract the public by its peculiarity and its distinctiveness from others.
- Authenticity—CAF can be confident that the management and employees are confident that the CSR concept is applied correctly in CAF's life.
- Transparency—willingness to provide information about itself, to allow independent assessment and social (political) control [28].
- Consistency in compliance with the CSR principles.

The interaction between all those affected by the organization (i.e., individuals, institutions, and other organizations—from here on "stakeholders") is largely determined by individual and institutional differences. In the broadest sense, the CAF stakeholder group includes military and civilian employees, partners, suppliers, the government and local government representatives, interest groups, the media, trade unions, and allied international military and civilian institutions and organizations [29,30].

A number of CSR theories emphasize that ensuring the smooth and efficient functioning of an organization is only possible if the needs of all stakeholders are adequately met [31–34]. Therefore, building a CSR in the CAF should include identifying the expectations and needs of key stakeholders and finding ways to align those needs, the needs of the members of CAF, and the needs and expectations of the CAF. This can be fulfilled if the CAF stakeholders share the values declared by CAF.

The specific character of the CAF within Czech society is codified in legislation which endows the CAF with the responsibility for the external safety of the state and, in cooperation with relevant institutions, internal safety and crisis management. Due to this specific set of responsibilities, members of the CAF, and especially military professionals, are held to high behavioral standards.

This special character of the military organization gives rise to contradictory requirements; on the one hand professional soldiers are trained, and thus expected, to be able to kill, while on the other they are forbidden to use these skills in their personal lives. This contradiction places both high requirements on a soldier's psychological resilience and moral compass, and emphasizes the importance of the long-term development of, and instruction in, ethical training for military professionals, for the sustainability of required ethical standards.

The values declared by CAF consist of a set of moral values, containing the basic moral values of the civil liberal-democratic society (the so-called moral minimum) and the specific values of the military profession. These values, which are defined in the Czech Army Doctrine [35], can be considered prudent in the military profession: Responsibility and a strong sense of duty, justice, dedication, courage, loyalty, and honor. For the focused and systematic development of ethical standards of CSR in the CAF, it is useful to examine the value orientation of its members and its transformation [36].

The authors focus their contribution to the social area of the social responsibility of the CAF, more specifically on the sustainable development of human capital in the area of professional ethics [37,38]. They assume that an important condition for the successful implementation of the CSR concept in the CAF environment [39] is the appropriate moral-value orientation of their members [40,41].

In connection with the preparation for service in the state security organization, i.e., a military career, the most frequently mentioned are high demands on the physical fitness of a soldier and his psychological resilience, i.e., the ability to cope with the burden [41]. The level of psychophysical fitness is grounded in the ability to efficiently handle changes and find a solution to tasks in a dynamically changing environment [42]. It is important to work with change, transformation, and the burden associated with it, such as coping with and realizing the change and the burden associated with it and the human system [43].

However, the military profession has an important value dimension [44]. The professional performance of the military profession requires the internalization of values characteristic of the military environment. Not only because it eases adaptation of the individual into this environment but also because the military profession is based on a high level of cooperation and emphasizes high standards of respect for professional morality [45,46]. Values that are accepted and acknowledged by an individual have a direct impact on their actions and subsequent group interactions [47,48]. Values and value attitudes affect, for example, an individual's ability to make decisions, formulate morally

substantiated claims, and act on them [49]. A relationship between values and moral reasoning can also be assumed [50,51]. Value attitudes are, in addition, an integral part of models of human behavior [52].

In the environment of the CAF, the sustainable development of human capital in the area of professional ethics must be supported by the introduction of standards for its measurement and objective assessment. However, these measurement and objective assessment standards are not set in the CAF environment; this manuscript responds to this gap and introduces the possibility [36]. Based on these findings, the degree of identification with the required values of the military profession can be predicted. Our results can be used for the sustainable development of human capital, especially in the area of moral value, in the preparation of the above-mentioned quality programs developing the organizational culture of the organization, and eliminating the likelihood of moral failure in the military profession (such as abuse of power, abuse of physical strength or special skills, e.g., to deal with personal situations) [53].

The current paper presents the first data collection of its kind. The authors aim to establish the value orientation of different sub-groups within the military and hope to identify how these groups differ in this respect. We hypothesize differences in value orientation related to groups of military rank (four groups: Higher officers, lower officers, non-commissioned officers and warrant officers, and students at the military university); the "rank" subgroup which an individual occupies, at least in part, shapes value orientation.

The main objective of this study was to find out whether value orientation changes according to the affiliation to the group of military rank. To meet the main objective, the following three research objectives were formulated: (1) To compare the measured value orientation based on respondents' military rank; (2) to determine whether the presence of lower-ranking individuals strengthens self-identification with the value "conformity." This value is comprising obedience, respect for authority, courtesy, and self-control; (3) to determine whether the presence of higher-ranking individuals strengthens self-identification with the value "power." This value comprises social power, authority, wealth, goodwill, and social recognition.

A high-level identification with the "power" value associated with a low-level identification with "conformity" can be a risk factor for CAF members, both with regard to stressful situations and the context of sustainable human capital development in the area of moral values. The resulting findings, which confirm that the value orientation of CAF officers varies depending on rank, are formulated in Section 5 of this article, along with recommendations for the sustainable development of CAF human capital in the area of ethical responsibility.

## 2. Materials and Methods

### 2.1. Research Objectives and Hypotheses

To meet the main objective (i.e., to determine if value orientation changes depending on the rank), the following three research objectives were formulated: (1) To compare the measured value orientation to respondents' ranks; (2) to determine whether lower ranks correlate with higher self-identification with "conformity," and (3) to find out whether rank is correlated with self-identification with "power."

To test the first sub-objective, the following set of null (H1–H10) and alternative (HA1–HA10) hypotheses (see Appendix A) were formulated:

**Hypothesis 1 (H1).** *The degree of universalism does not depend on military rank.*

**Hypothesis 2 (H2).** *The degree of benevolence does not depend on membership of the military rank.*

**Hypothesis 3 (H3).** *The degree of conformism does not depend on membership of the military rank.*

**Hypothesis 4 (H4).** *The importance of tradition does not depend on membership of the military rank.*

**Hypothesis 5 (H5).** *The importance of security does not depend on membership of the military rank.*

**Hypothesis 6 (H6).** *The importance of power does not depend on membership of the military rank.*

**Hypothesis 7 (H7).** *The importance of success does not depend on membership of the military rank.*

**Hypothesis 8 (H8).** *The degree of indulgence does not depend on membership of the military rank.*

**Hypothesis 9 (H9).** *The importance of stimulation does not depend on membership of the military rank.*

**Hypothesis 10 (H10).** *The degree of independence does not depend on membership of the military rank.*

The hypotheses H3 and HA3 were used to address the second sub-objective as well, and the hypotheses H10 and HA10 were used to address the third sub-objective.

*2.2. Methods and Tools of Data Collection and Evaluation*

Shalom Schwartz's approach became the starting point for identifying value profiles. The theoretical reason is Schwartz's holistic concept, which includes not only the psychological interpretation of values but also the possibility of transferring aggregate results to the group level [45]. The advantage of Schwartz's approach is that it tends to identify the most significant values and are applied in various life situations [47,54,55].

Schwartz' values [56–59] have five formal characteristics: These are concepts or beliefs that relate to desirable end states or behaviors, transcend specific situations, control the selection or evaluation of behavior and phenomena, are arranged according to their relative importance, and do not stand alone but rather form a structure with relationships, similarities, and oppositions. It is assumed that the entire value structure of the respondents is measured, i.e., that no significant value dimension is omitted [60].

To determine the value orientation of CAF members, Schwartz's Portrait Values Questionnaire (PVQ) was used in a reduced version with 21 entries. This test focusses on 10 basic value types: Security (self-security: Harmony and stability of the individual and the security of society and relationships, conformism, and tradition); conformity (self-control or self-discipline in action, behavior, etc., wherever there is a possibility that an individual's activity can interfere with or endanger others and violate society's expectations and standards, respect for authority); tradition (respect, devotion, and acceptance of customs and ideas, which traditions or religion offer, modesty, acceptance of devotion, honoring traditions); independence/self-direction (independence, freedom in behavior, creativity, curiosity, need for autonomy, independence, and control); stimulation (variability, stimulation, newness, life challenges); hedonism/self-indulgence (joy, pleasure and sensual enjoyment, self-satisfaction, need for enjoyment, pleasure, delight); power (influence, power, social status, prestige, control or domination over people and resources); success/achievement (success, recognition, personal performance, personal success based on demonstrating abilities compared to others); universalism (justice, respect, tolerance and care for the good of all people and the whole of nature, social justice, equality, peace around the world, environmental protection), and benevolence (the benefit of the people we are in everyday contact, protecting and enhancing the prosperity of people we often encounter, friendship, love, loyalty, and benevolence). Each of the 21 PVQ items is characterized by a portrait characterizing the type of personality attitude [61–63].

These are grouped into four higher-order value types: Conservatism, openness to change, ego-enhancement, and overcoming yourself. The four higher order types of values are characterized as follows: Conservatism is made up of the values of security, conformity, and tradition. Openness to change is formed by the values of self-direction, stimulation, and hedonism/self-indulgence. Ego-enhancement is saturated with success and power. Overcoming yourself is fulfilled by benevolence and universalism [60,62,63].

Studies of the European Social Survey have confirmed the hypothesis of the existence of another two higher dimensions of values oriented to the individual, represented in blue (Figure 1) (power, success/achievement, hedonism/self-indulgence, stimulation, and self-direction) or to collective interests, represented by the yellow part (universalism, benevolence, tradition, conformity, and security) [48,54,60,64,65].

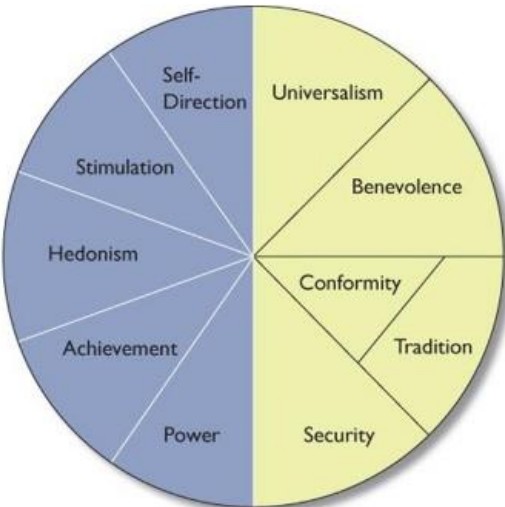

**Figure 1.** The structure of human values according to Shalom Schwartz. Source: https://www.slideshare.net/jonrwallace/values-11550083.

The reliability coefficient of Cronbach's internal consistency of the PVQ tools is sufficient, based on ESS European Social Data Analysis, which estimates the reliability of the PVQ tool in the Czech Republic between 0.67 to 0.75 [47].

The Kruskal–Wallis test was used to compare value scores military rank because the assumption of normality was violated by all values according to the Shapiro-Wilk test. If dependency was demonstrated, post hoc tests were performed based on multiple average ranking comparisons. The mean and standard deviation were calculated for the comparison groups and the median, lower, and upper quartile, minimum and maximum were shown in a box graph. All calculations were performed using STATISTICA CZ 12, the significance level was 0.05.

## 2.3. Research Sample and Research Process

For this experiment, 40 non-commissioned and warrant officers (NCO and WO), 171 cadets—students at the University of Defense (S), 40 lower-ranking commissioned officers (LCO), and 35 higher-ranking commissioned officers (HCO) were recruited. All participants were between 20 and 55 years of age and the sample was generated through random stratified sampling. The sample was stratified according to the above-mentioned categories and age and gender were not taken into account. This experiment included 286 participants. The return rate of questionnaires was 100%. Although participants were not equally distributed across test-groups, their number per test-group was sufficient to assume that these results reflect a general trend.

Cadets were presented with the PVQ as part of one of their classes while the other participants filled it out during training sessions. Test-group size varied from 20 to 40 individuals and it took participants approximately 10 min to complete the task.

Respondents received a form with statements marked A to U and assigned one of the following options for each item: (1) I am very similar; (2) I am similar; (3) I am rather similar; (4) I am rather not similar; (5) I am not similar; (6) I am not similar at all. In the PVQ form, versions for men and women were merged into one form.

Respondents were guaranteed anonymity. They were instructed and stressed the need to respond to each item as honestly as possible, not with regard to the expected solution. During the administration of the questionnaire, an authorized person was present who was able to answer any questions of the respondents in order to eliminate any misunderstandings.

## 3. Results

### 3.1. First Sub-Objective Solution

The following results were measured for the first sub-objective (Table 1):

**Table 1.** Kruskal–Wallis test and numerical characteristics for values.

| Values | NCO and WO | | LCO | | HCO | | Students | | *p*-Value | H$_0$ Decision |
|---|---|---|---|---|---|---|---|---|---|---|
| | AV * | SD ** | AV | SD | AV | SD | AV | SD | | |
| Universalism | 2.77 | 0.94 | 2.26 | 0.78 | 2.46 | 0.61 | 2.56 | 0.81 | 0.057 | not reject |
| Benevolence | 2.48 | 0.78 | 2.14 | 0.77 | 2.31 | 0.54 | 2.03 | 0.72 | 0.001 | reject |
| Conformity | 3.21 | 1.05 | 2.50 | 0.97 | 2.52 | 0.81 | 3.28 | 1.01 | 0.000 | reject |
| Tradition | 3.14 | 1.06 | 2.51 | 0.92 | 2.41 | 0.68 | 3.01 | 1.06 | 0.001 | reject |
| Security | 2.95 | 0.94 | 2.34 | 1.11 | 2.80 | 0.83 | 3.04 | 0.94 | 0.004 | reject |
| Success | 3.28 | 0.97 | 3.54 | 1.17 | 3.30 | 1.14 | 2.90 | 1.09 | 0.005 | reject |
| Hedonism | 2.30 | 0.80 | 3.03 | 1.14 | 3.00 | 0.82 | 2.27 | 1.03 | 0.000 | reject |
| Stimulation | 2.60 | 1.19 | 3.31 | 1.23 | 3.34 | 0.95 | 2.46 | 0.92 | 0.000 | reject |
| Self-direction | 2.28 | 0.80 | 2.16 | 0.66 | 2.17 | 0.63 | 2.32 | 0.76 | 0.596 | not reject |
| Power | 3.08 | 1.08 | 3.56 | 1.06 | 3.20 | 0.93 | 3.24 | 0.88 | 0.333 | not reject |

AV *—average, SD **—standard deviation.

For universalism, the *p*-value (the smallest level of significance at which the null hypothesis can still be rejected) was based on the Kruskal–Wallis test with respect to three decimal places 0.057, i.e., higher than 0.05. The Null Hypothesis H1 was not rejected. The level of significance of 0.05 did not prove the dependence of the degree of universalism on military rank. The medians, lower and upper quartiles, minimal and maximal limits are shown by means of a box graph (Figure A1).

For the value of benevolence, the *p*-value of the Kruskal–Wallis test (Table 1.) with respect to three decimal places was 0.001, i.e., less than 0.05. The Null Hypothesis H2 was rejected in favor of an alternative hypothesis HA2. The level of significance of 0.05 did not prove the dependence of the degree of benevolence on military rank. Medians, lower and upper quartiles, minimal and maximal limits are shown by means of a box graph (Figure A2).

While measuring the value of benevolence, all pairs of the groups were then compared on the basis of post hoc tests based on multiple average ranking comparisons. Their resulting *p*-values (adjusted according to Bonferroni correction) are shown in the following Table 2.

**Table 2.** Post hoc tests—*p*-values—table benevolence.

| Group | HCO | LCO | NCO and WO | Students |
|---|---|---|---|---|
| **HCO** | - | 0.835 | 1.000 | 0.056 |
| **LCO** | 0.835 | - | 0.212 | 1.000 |
| **NCO and WO** | 1.000 | 0.212 | - | 0.002 |
| **Students** | 0.056 | 1.000 | 0.002 | - |

For conformity, the *p*-value of the Kruskal–Wallis test with respect to three decimal places was 0.001, i.e., less than 0.05. The Null Hypothesis H$_{03}$ was rejected in favor of an alternative hypothesis HA3. The level of significance of 0.05 did not prove the dependence of the degree of conformity on military rank.

While measuring conformity, all pairs of the groups were then compared on the basis of post hoc tests based on multiple average ranking comparisons, their resulting *p*-values are shown in the following Table 3.

**Table 3.** Post hoc tests—*p*-values—table conformity.

| Group | HCO | LCO | NCO and WO | Students |
|---|---|---|---|---|
| HCO | - | 1.000 | 0.021 | 0.000 |
| LCO | 1.000 | - | 0.028 | 0.001 |
| NCO and WO | 0.021 | 0.028 | - | 1.000 |
| Students | 0.000 | 0.001 | 1.000 | - |

At a significance level of 0.05, a difference in the degree of conformity between students and lower-commissioned officers ($p = 0.001$), students and higher-commissioned officers ($p = 0.000$), non-commissioned officers and warrant officers and higher-commissioned officers ($p = 0.021$), non-commissioned officers and warrant officers and lower-commissioned officers ($p = 0.028$) was shown. The degree of conformity of non-commissioned officers and warrant officers was statistically significantly higher than that of higher- and lower-commissioned officers, and the degree of the conformity in students was statistically significantly higher than that of higher- and lower-commissioned officers. No statistically significant differences were found between the other pairs of groups. Medians, lower and upper quartiles, minimal and maximal limits are shown by means of a box graph (Figure A3).

For the value of tradition, the *p*-value of the Kruskal–Wallis test was taken with respect to three decimal places of 0.001, i.e., less than 0.05. The Null Hypothesis $H_{04}$ was rejected in favor of an alternative hypothesis $H_{A4}$. The significance level of 0.05 demonstrates a dependence of self-identification with tradition is predicted by military rank.

While measuring the value of tradition, all pairs of the groups were then compared on the basis of post hoc tests based on multiple average ranking comparisons, their resulting *p*-values are shown in the following Table 4.

**Table 4.** Post hoc tests—*p*-values—table tradition.

| Group | HCO | LCO | NCO and WO | Students |
|---|---|---|---|---|
| HCO | - | 1.000 | 0.010 | 0.017 |
| LCO | 1.000 | - | 0.033 | 0.063 |
| NCO and WO | 0.010 | 0.033 | - | 1.000 |
| Students | 0.017 | 0.063 | 1.000 | - |

At a significance level of 0.05, there was a difference in the importance of tradition between students and higher-commissioned officers ($p = 0.017$), between non-commissioned officers and warrant officers and higher-commissioned officers ($p = 0.010$), and between non-commissioned and warrant officers and lower-commissioned officers ($p = 0.033$). The importance of tradition was statistically significantly higher for students than for higher-commissioned officers, and for non-commissioned officers and warrant officers statistically significantly higher than for lower- and higher-commissioned officers. No statistically significant differences were found between the other pairs of groups. Medians, lower and upper quartiles, minimal and maximal limits are shown by means of a box graph (Figure A4).

For the value of security, *p*-value of the Kruskal–Wallis test based on the three decimal places was 0.004, i.e., lower than 0.05. The Null Hypothesis $H_{05}$ was rejected in favor of an alternative hypothesis $H_{A5}$. At a significance level of 0.05 was proven dependence of safety importance on belonging to the rank corps.

While measuring the value of security, all pairs of the groups were then compared on the basis of post hoc tests based on multiple average ranking comparisons, their resulting *p*-values are shown in the following Table 5.

**Table 5.** Post hoc tests—*p*-values—table security.

| Group | HCO | LCO | NCO and WO | Students |
|---|---|---|---|---|
| HCO | - | 0.504 | 1.000 | 1.000 |
| LCO | 0.504 | - | 0.059 | 0.003 |
| NCO and WO | 1.000 | 0.059 | - | 1.000 |
| Students | 1.000 | 0.003 | 1.000 | - |

At a significance level of 0.05, there was a difference in the importance of security between students and lower-commissioned officers ($p = 0.003$). For students, security was statistically more important than for lower-commissioned officers. No statistically significant differences were found between the other pairs of groups. Medians, lower and upper quartiles, minimal and maximal limits are shown by means of a box graph (Figure A5).

For the value of success, the *p*-value of the Kruskal–Wallis test was taken with respect to three decimal places of 0.005, i.e., less than 0.05. The Null Hypothesis H6 was rejected in favor of an alternative hypothesis HA6. At a significance level of 0.05, there was shown the dependence of the importance of success on the membership of the military rank.

While measuring the value of success all pairs of the groups were then compared on the basis of post hoc tests based on multiple average ranking comparisons, their resulting *p*-values are shown in the following Table 6.

**Table 6.** Post hoc tests—*p*-values—table success.

| | HCO | LCO | NCO and WO | Students |
|---|---|---|---|---|
| HCO | - | 1.000 | 1.000 | 0.491 |
| LCO | 1.000 | - | 1.000 | 0.017 |
| NCO and WO | 1.000 | 1.000 | - | 0.211 |
| Students | 0.491 | 0.017 | 0.211 | - |

At a significance level of 0.05, there was a difference in the importance of success between students and lower-commissioned officers ($p = 0.017$). The importance of success was significantly lower for students than for lower-commissioned officers. No statistically significant differences were found between the other pairs of groups. Medians, lower and upper quartiles, minimal and maximal limits are shown by means of a box graph (Figure A6).

For the value of hedonism/self-indulgence the *p*-value of the Kruskal–Wallis test was taken with respect to three decimal places of 0.000, i.e., less than 0.05. The Null Hypothesis H7 was rejected in favor of an alternative hypothesis HA7. The significance level of 0.05 showed the dependence of the degree of hedonism/self-indulgence on the membership of the military rank.

While measuring the value of hedonism/self-indulgence, all pairs of the groups were then compared on the basis of post hoc tests based on multiple average ranking comparisons, their resulting *p*-values are shown in the following Table 7.

**Table 7.** Post hoc tests—*p*-values—table hedonism/self-indulgence.

| Group | HCO | LCO | NCO and WO | Students |
|---|---|---|---|---|
| HCO | - | 1.000 | 0.007 | 0.000 |
| LCO | 1.000 | - | 0.029 | 0.001 |
| NCO and WO | 0.007 | 0.029 | - | 1.000 |
| Students | 0.000 | 0.001 | 1.000 | - |

At a significance level of 0.05, a difference in hedonism/self-indulgence was shown between students and higher-commissioned officers ($p = 0.000$), students and lower-commissioned officers

($p$ = 0.001), non-commissioned officers and warrant officers and higher-commissioned officers (0.007), and non-commissioned officers and warrant officers and lower-commissioned officers (0.029). The degree of hedonism/self-indulgence was statistically significantly lower for students than for higher- and lower-commissioned officers, and for non-commissioned officers and warrant officers statistically significantly lower than for higher- and lower-commissioned officers. No statistically significant differences were found between the other pairs of groups. Medians, lower and upper quartiles, minimal and maximal limits are shown by means of a box graph (Figure A7).

For the value of stimulation, the $p$-value of the Kruskal–Wallis test was taken with respect to three decimal places of 0.000, i.e., less than 0.05. The Null Hypothesis H8 was rejected in favor of an alternative hypothesis HA7. The significance level of 0.05 showed the dependence of the importance of stimulation on the military rank.

While measuring the value of stimulation, all pairs of the groups were then compared on the basis of post hoc tests based on multiple average ranking comparisons, their resulting $p$-values are shown in the following Table 8.

**Table 8.** Post hoc tests—$p$-values—table stimulation.

| Group | HCO | LCO | NCO and WO | Students |
|---|---|---|---|---|
| **HCO** | - | 1.000 | 0.008 | 0.000 |
| **LCO** | 1.000 | - | 0.043 | 0.001 |
| **NCO and WO** | 0.008 | 0.043 | - | 1.000 |
| **Students** | 0.000 | 0.001 | 1.000 | - |

At a significance level of 0.05, there was a difference in the importance of stimulation between students and higher-commissioned officers ($p$ = 0.000), students and lower-commissioned officers ($p$ = 0.001), non-commissioned officers and warrant officers and higher-commissioned officers (0.008), and non-commissioned officers and warrant officers and lower-commissioned officers (0.043). The importance of stimulation was statistically significantly lower for students than for lower- and higher-commissioned officers, and for non-commissioned officers and warrant officers statistically significantly lower than for lower- and higher-commissioned officers. No statistically significant differences were found between the other pairs of groups. Medians, lower and upper quartiles, minimal and maximal limits are shown by means of a box graph (Figure A8).

For the value of self-direction, the $p$-value of the Kruskal–Wallis test was taken with respect to three decimal places of 0.596, i.e., higher than 0.05. The Null Hypothesis $H_{09}$ was not rejected. The level of significance of 0.05 did not prove the dependence of the degree of independence on the membership of the rank corps. Medians, lower and upper quartiles, minimal and maximal limits are shown by means of a box graph (Figure A9).

For the value of power, the $p$-value of the Kruskal–Wallis test based on three decimal places was 0.333, i.e., higher than 0.05. The Null Hypothesis $H_{010}$ was not rejected. The significance level of 0.05 did not prove the dependence of the importance of power on the rank of corps. Medians, lower and upper quartiles, minimal and maximal limits are shown by means of a box graph (Figure A10).

Individual findings confirmed that the values of benevolence, conformity, tradition, security, success, hedonism/self-indulgence, and stimulation showed the influence of rank corps on value orientation. The values of universalism, self-direction, and power did not demonstrate the influence of corps membership on the value orientation. It can, therefore, be confirmed that the value orientation of the CAF members varies depending on the military rank groups.

*3.2. Second Sub-Objective Solution*

Based on the results of the Kruskal–Wallis test it was shown that the level of significance ($p$-value) for the conformity value is 0.000, i.e., less than 0.05 (see. Table 1). The $H_{03}$ hypothesis was rejected in

favor of the $H_{A3}$ hypothesis, which was accepted. The dependence of the degree of conformity on membership of the military rank groups was proved.

Multiple comparisons of the study groups (students, non-commissioned officers, lower-commissioned officers, higher-commissioned officers) using post hoc tests showed differences in the degree of conformity at the significance level of 0.05 (see Table 3) between students and lower-commissioned officers ($p = 0.001$), students and higher-commissioned officers ($p = 0.000$), between non-commissioned officers and warrant officers and higher-commissioned officers ($p = 0.021$), and between non-commissioned officers and warrant officers and lower-commissioned officers ($p = 0.028$). The conformism rate of non-commissioned officers and warrant officers and students was statistically significantly higher than that of higher- and lower-commissioned officers (see Figure A3). It was confirmed that inclusion in the lower military rank groups (students, non-commissioned officers, and warrant officers) strengthens identification with the value of conformity (obedience, respect for authorities, politeness, and self-control).

### 3.3. Third Sub-Objective Solution

The null hypothesis $H_{010}$ was used to address the third sub-objective, "to find out whether the inclusion in the lower ranking strengthens the identification with the power, which is saturated with social power, authority, wealth, goodwill, and social recognition." The degree of power does not depend on membership of the rank corps and the alternative hypothesis HA10: The degree of power depends on membership of the rank corps. Based on the results of the Kruskal–Wallis test it was shown that the level of significance ($p$-value) H10 is 0.333, i.e., higher than 0.05 (see Table 1). The hypothesis $H_{010}$ was not rejected. The dependence of the degree of power on the membership of the military rank groups was not proved.

### 4. Discussion

Research on value orientation in the environment of the state security organizations is not a subject of systematic interest. The authors could use partial knowledge from a very limited number of foreign research studies on moral-value orientation in the military, police, and security environment [66–74].

For the effective development and integration of CSR in the CAF it would be desirable and expected that the degree of identification with the values of security, conformity, and tradition would increase with rank. The security value, which in the military-professional context can be interpreted as a desire for security and a secure and stable society [47,54,60], was most preferred by students and the biggest difference in preferences was between students and lower-commissioned officers. The degree of conformism, which can be interpreted in a military-professional context as a self-discipline in the exercise of the military profession, obedience, courtesy, compliance with standards and regulations [48,54,60], was significantly higher for non-commissioned officers and warrant officers and students than for higher- and lower-commissioned officers. Our data also indicate that the presence of lower-ranking individuals (students, non-commissioned officers, and warrant officers) raises self-identification with this value. The value of tradition, which in a military-professional context can be interpreted as respect, loyalty, and loyalty to a military institution, acceptance of customs, ideals, rituals and ceremonies associated with the military culture [48,54,60], has shown a difference in the importance of the perception of tradition between students and non-commission officers and warrant officers on the one hand and lower- and higher-commissioned officers on the other. The importance of tradition was significantly higher for students and non-commissioned officers and warrant officers than for lower- and higher-commissioned officers. For other values, on the other hand, a lower degree of dependence is desirable for the identification with the given value on the affiliation to a higher military rank. It is primarily power, which in the military-professional context can be interpreted as an effort to control the personnel and resources of the military institution and to build a dominant position in its structure, furthermore, the values of success, which in the military-professional context can be interpreted as the pursuit of a successful career, the achievement of the highest possible rank and

service status, and the corresponding professional and social status, and hedonism/self-indulgence, which in the military-professional context can be interpreted as an effort to satisfy sensory needs through military activities.

To stimulate values, which, in the military-professional context can be interpreted as the need to successfully face professional challenges and seek new solutions and approaches, self-direction, which in the military-professional context can be interpreted as an effort for autonomy in thinking, decision-making and action despite inclusion in the military structure, curiosity and creativity; universalism, which in the military-professional context can be perceived as the respect for human life and the effort to sensitively treat human and material resources; and benevolence, which can be interpreted in the military-professional environment as the support for superiors and collaborators and the development of subordinates [48,54,60]—it is desirable that they apply across the value spectrum. For three values, namely universalism, self-direction, and power, the influence of military rank groups was not confirmed, so there is no demonstrable risk of linking a high rate of power with a low rate of conformism in any ranks.

For the value of success, its importance was perceived as significantly lower among students (compared to other ranks). In hedonism/self-indulgence, a difference in the degree of hedonism/self-indulgence was confirmed between students and non-commissioned and warrant officers and higher- and lower-commissioned officers. The degree of hedonism/self-indulgence was significantly lower for students and non-commissioned officers than for higher- and lower-commissioned officers. The value of stimulation showed a difference in the perception of the importance of stimulation between students and non-commissioned officers and warrant officers on the one hand, and higher- and lower-commissioned officers on the other. Stimulation was perceived by students and non-commissioned officers and warrant officers as significantly less important than by higher and lower officers. The value of benevolence showed a difference in the rate of benevolence between students and non-commissioned officers and warrant officers. The benevolence rate of non-commissioned and warrant officers was statistically significantly higher than that of students.

These findings also suggest that for the sustainable development of the social field of CSR in the CAF, specifically the sustainable development of human capital with regard to the professional ethics and moral value orientation of CAF members, it is necessary to take into account the individual specifics for the military rank groups (education, professional experience, professional requirements placed on them, inclusion in the CAF hierarchy, etc.). In addition, it is necessary to take into account the value orientation of Czech society. Currently, the Czech population, compared to the European standard, is characterized by a lower preference for openness to change (based on the values of self-direction, stimulation, and hedonism/self-indulgence) [48,60–64] and a larger preference of conservatism (based on the values of security, conformity, and tradition) [48,60–64].

The discussion above also leads to recommendations for further research, which must be concentrated in three directions: Firstly, future expansions on this should include more ranks within the CAF to see in how far these findings hold up for the whole organization. Secondly, a focus on senior officers in combination with refined research tools is advised. And the third direction is a comparative research of value orientation of the Czech military environment within the international military environment.

## 5. Conclusions

The aim of the study was to find out whether and how the value orientation of CAF members changes depending on their military rank in order to predict the need for, and focus of, educational activities of personnel in the area of professional ethics and to ensure the social pillar for its socially responsible sustainable development.

In relation to the research studies the topic of "How much time is necessary for a CAF member to identify with the values of the military profession and organization?" is often raised in discussions in the Czech military environment. These results suggest that it is not only about time but also—and

more importantly—about how the military environment is involved in instilling desirable values in personnel throughout the military career. The results of foreign research can be used only rarely in this context, as the socio-cultural environment may have a significant influence on the preference of some values [50].

Our data indicate that the value orientation of CAF members changes depending on their military rank. This has been demonstrated for seven out of 10 values forming a value portrait according to Schwartz's theory [53,56,58–61,65], namely security, conformism, tradition, stimulation, hedonism/self-indulgence/success, and benevolence.

Based on our findings, follow-up recommendations for the sustainable development of the social pillar of social responsibility, and human capital in the area of professional ethics can be made. The identification and evaluation of value profiles have been shown to yield important insights into the moral-value orientation of CAF members. The authors, therefore, recommend to continue to monitor the value profiles of CAF members of all ranks in different stages of their careers. It is only on the basis of a greater number of repeatedly obtained data that it is possible to objectively create a stable personality-value profile of individuals and ranks and to assess the degree of identification with the required values of the military profession (in accordance with the emphases of CAF Doctrine) for the respective military ranks. Such data could also be used for the sustainable development of human capital in the form of individual and professional counselling.

The second recommendation is to focus attention not only on students and members of lower ranks but on the preparation of ethical education/leadership programs aimed at developing the moral-value potential of CAF officers and establishing the foundations of sustainable moral competence for the military profession in the context of CSR.

To develop these programs, it is necessary to look for a systematic solution which allows for the creation of a system of ethics education, training, and ethical guidance in the CAF context. This involves educational institutions and stakeholders as well as the necessary prerequisites for its creation. The preparation of the particular programs should accept the results of above-described research and, in the future, to use the knowledge and results of further (future) research of the same or extension type.

Based on these findings, it seems optimal to create a separate program for students at the University of Defense, for non-commissioned officers and a warrant officer and for higher- and lower-commissioned officers. These programs must accentuate the findings on the value preferences of the individual ranks. The specific design of a system of ethical education, training, and leadership in the CAF as well as the content design of the individual programs will, due to their complexity and comprehensiveness, be the subject of further publications by the authors.

**Author Contributions:** Conceptualization, Z.M. and I.N.; methodology, I.N.; validation, J.F., formal analysis, J.F.; resources, Z.M.; data curation, I.N.; writing—original draft preparation, Z.M.; writing—review and editing, I.N. and J.F. All authors have read and agreed to the published version of the manuscript.

**Funding:** This research received no external funding.

**Conflicts of Interest:** The authors declare no conflict of interest.

## Appendix A

**Hypothesis A1 (HA1).** *The degree of universalism depends on membership of the military rank.*

**Hypothesis A2 (HA2).** *The degree of benevolence depends on membership of the military rank.*

**Hypothesis A3 (HA3).** *The degree of conformism depends on membership of the military rank.*

**Hypothesis A4 (HA4).** *The importance of tradition depends on membership of the military rank.*

**Hypothesis A5 (HA5).** *The importance of security depends on the military rank.*

**Hypothesis A6 (HA6).** *The importance of power depends on membership of the military rank.*

**Hypothesis A7 (HA7).** *The importance of success depends on membership of the military rank.*

**Hypothesis A8 (HA8).** *The degree of indulgence depends on membership of the military rank.*

**Hypothesis A9 (HA9).** *The importance of stimulation depends on membership of the military rank.*

**Hypothesis A10 (HA10).** *The degree of independence depends on membership of the military rank.*

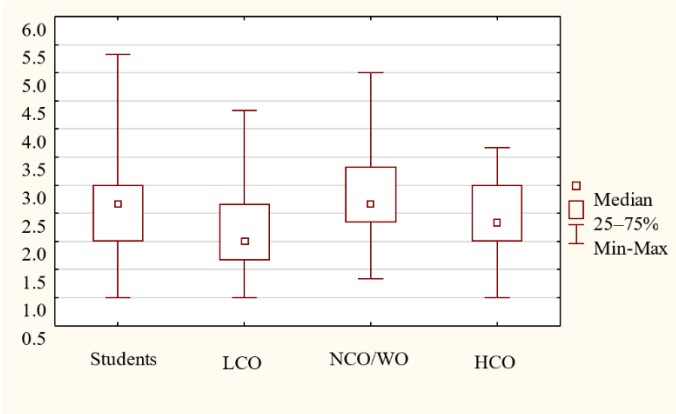

**Figure A1.** Boxplot for Universalism.

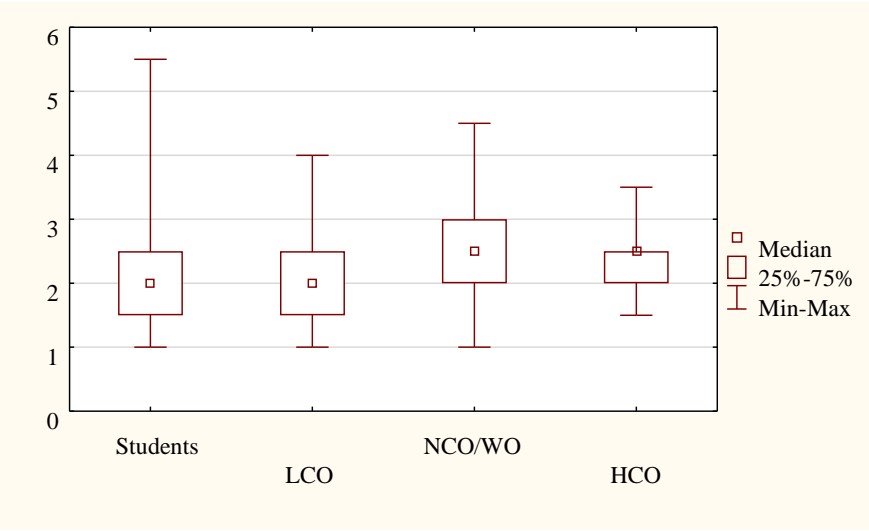

**Figure A2.** Boxplot for Benevolence.

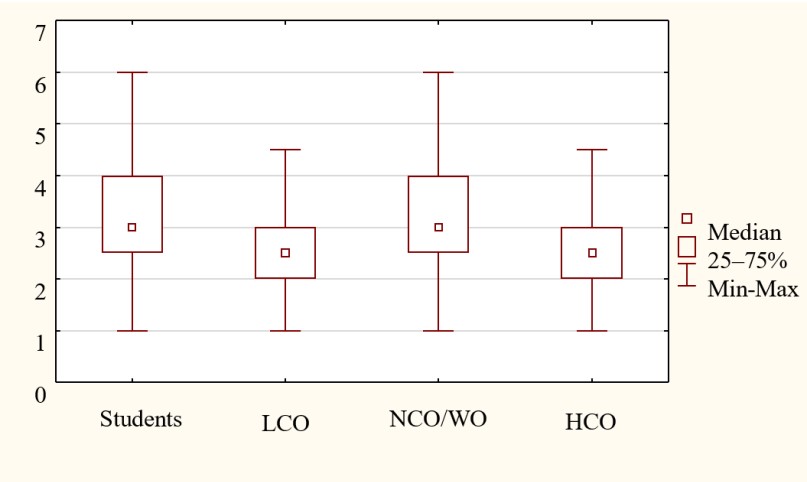

**Figure A3.** Boxplot for Conformity.

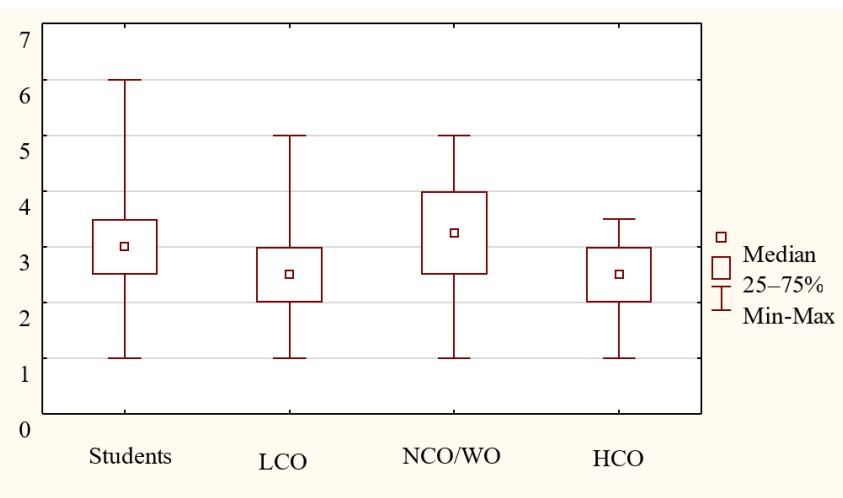

**Figure A4.** Boxplot for Tradition.

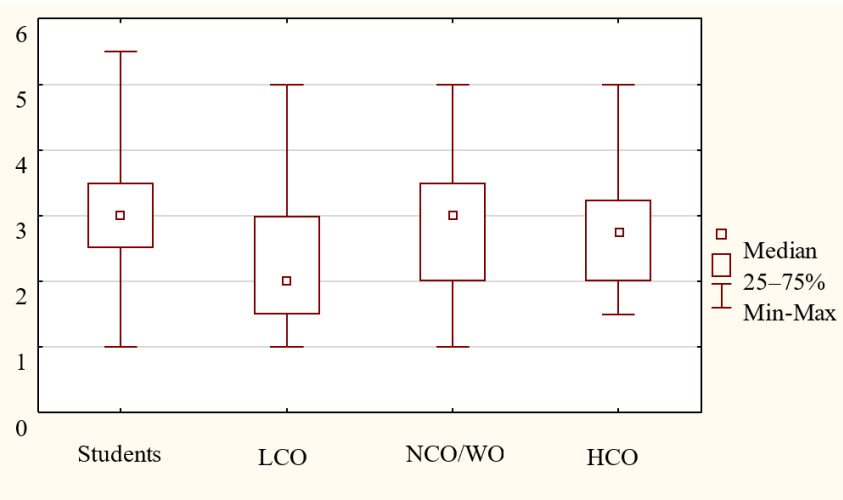

**Figure A5.** Boxplot for Security.

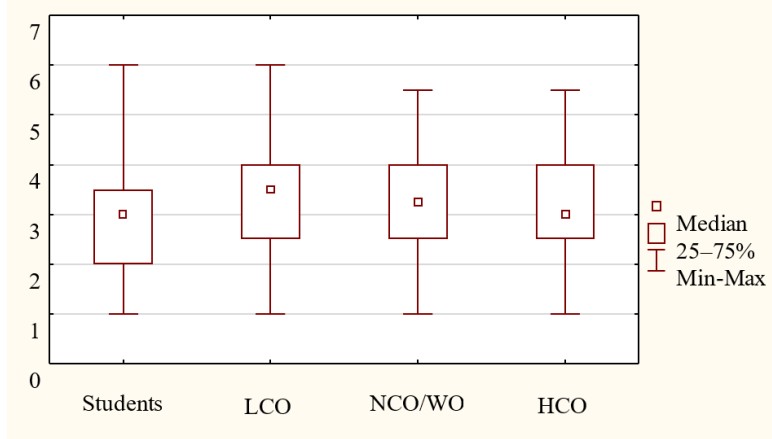

**Figure A6.** Boxplot for Success.

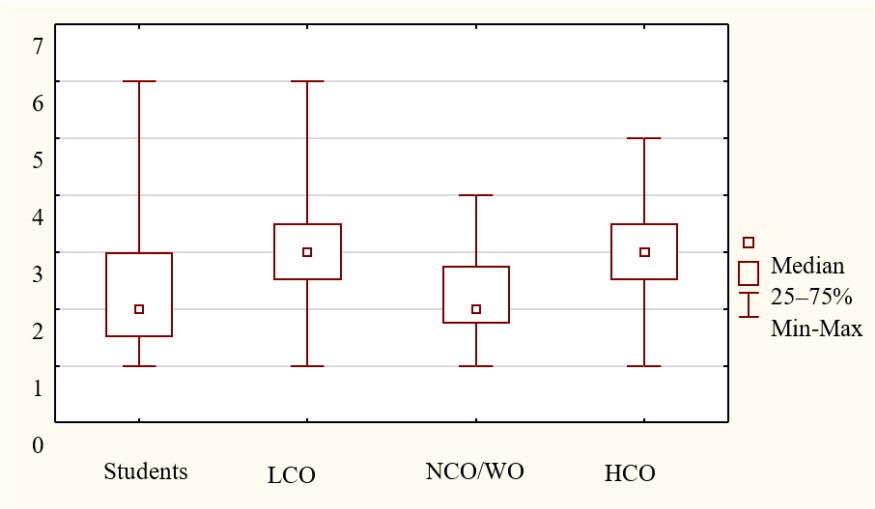

**Figure A7.** Boxplot for Hedonism/Self-indulgence.

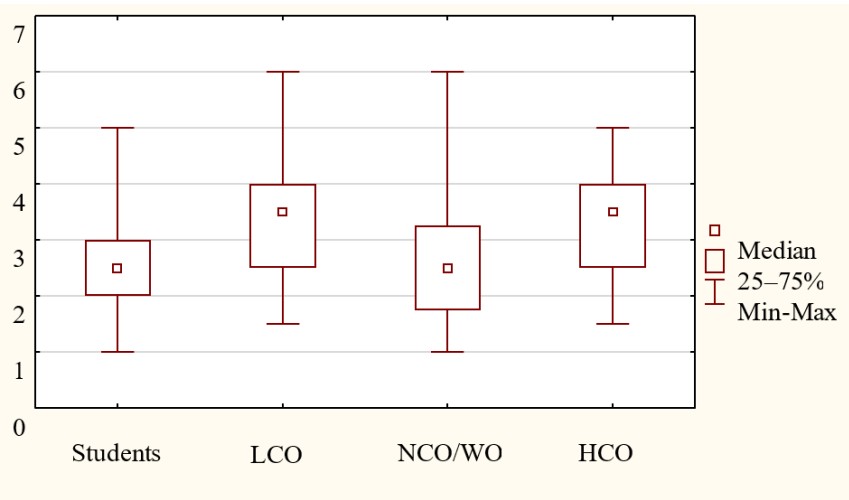

**Figure A8.** Boxplot for Stimulation.

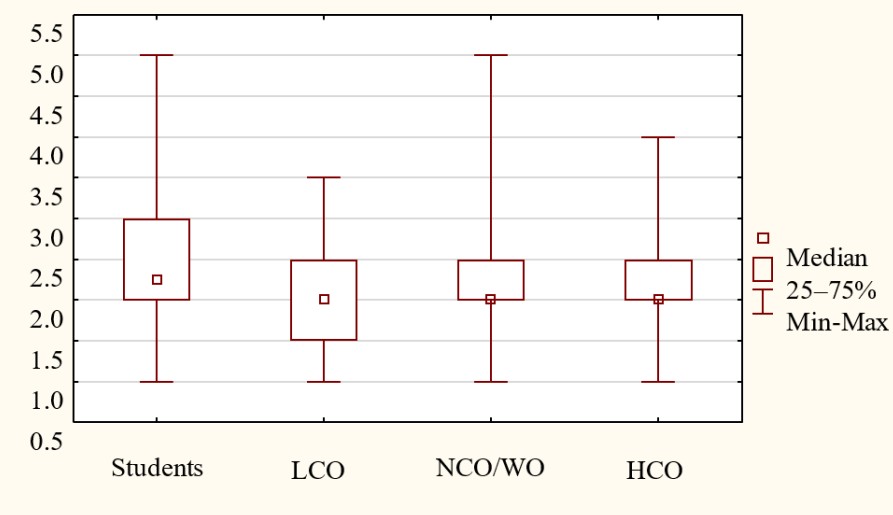

**Figure A9.** Boxplot for Self-direction.

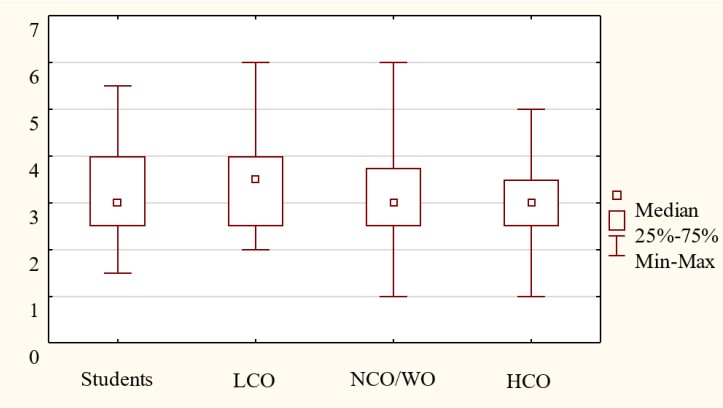

**Figure A10.** Boxplot for Power.

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
