# Peer review of "The Moral-Value Orientation—A Prerequisite for Sustainable Development of the Corporate Social Responsibility of a Security Organization"

_sustainability, doi:10.3390/su12145718_

Round 1

Reviewer 1 Report

The main problem I see is the missing CSR dimension. What I mean by that. The CSR element appears just as introductory element and is almost lost in the whole rest of the article. The article would be better red and understood, if it would focus just on value or human resource management issues, sustainable human resource management/development. The article is interesting and good (need for work on it still), but as the core element of CSR, sustainability is missing, I think. In now, as coming from sustainability perspective, we can link almost everything to it, however... I would still suggest to find another journal for its publication.

Also, the article need an English language review (native speaker's review).

Review also the formulation of hypotheses. The word "depend" is used, however is it really dependability type relationship defined? Please reconsider.

Reviewer 2 Report

Manuscript Title: The moral-value orientation - a prerequisite for sustainable development of the Corporate Social Responsibility of a security organization.

The formulation of the research gap is not conceived. Please develop a better grounding of the research problem. What do we already know? What is it that we do not know? Why do we need to know this and why is this important? But perhaps more so, what is the value of this paper? Both theoretically and managerially. Who is the audience of this paper? When you write up the paper again, please consider this.

The literature review reads more like a list of previous research on a variety of topics rather than a theory section that explains how your different concepts are related. Try to integrate this section better and build a stronger case of the need for your study.

The authors failed to justified the hypothesis.  

The discussion part is too long. It should be specific. 

The conclusion section is too short. It generally focuses on the study recommendations. Line539. Don't mix up study limitations with discussion.

Reviewer 3 Report

It was a very interesting paper.

Author Response

Thank you for your Review Report.

Reviewer 4 Report

Dear Authors

Thank you for contributing to the CSR domain by writing this article. I think you dove into CSR literature and applied it to a fairly original organization. Your introduction and problem definition is interesting. You provide an overview of the meaning of CSR, sustainability and how it can be managed by taking into account multiple stakeholder's visions and opinions. You even mention the importance of its strategic integration, instead of using it as PR.

However, the reader does not get a good understanding of the 'case' you use to further develop your  research subject (the organization). What is their strategy, what is their current tactic and how is CSR really incorporated? Which values do they adhere? 

After this introduction the reader  get a bit lost. You write about management, CSR, stakeholders .. and then you make a big leap to 'values'. If you want to look at values ​​at a strategic level, you have to look at the mission, vision ... of the organization. What values ​​do they represent?

Next, you need to deepen the gap between these values ​​and how each individual in the organization may or may not have this value orientation. (references, literature)

Then you have to prove why people at different levels of the organization should or should not have certain values. (references, literature)

If you work out the subject step by step, it can become understandable why your research is going into 'values' of different individuals in the organization. But in general, it's a big way from your introduction to your own research.

If you choose to use Schwartz (there may be more 'MORAL' oriented frameworks, as your title suggests what the paper is about), better explain the relationship between the individual value level and the organization's value orientation. Schwartz describes and maesures values as rather stable drivers (an orientation) for ones live and one's choices. What does that mean for what you are interested in. Should the organization recruit people based on a value orientation? 

The framework is well explained, but you put it under methodology, although it is the core of your contribution as suggeste in the title and in teh research part. 

You can explain the sample better .. how many respondents in each category, quota, layered, snowball .. How did you reach them ... What does 278 mean .. is it representative, how reliable is it .. Why didn't you use the original Schwartz scale (-1 ... +7)? Salomon Schwartz had reasons for including a negative answer possibility. Did you formulate the question as it is meant to be? 

The results can be explained better (more to the point, more accurately). Try to summarize more efficiently what you have found. Your discussion is not really related to your literature to your problem definition nether to  the long introduction of the paper. This can be improved.

I wish you good luck in your research. 

Round 2

Reviewer 1 Report

Comments:

  1. The "dot" after the title is not needed.
  2. Abstract and the Title. The content in the title and the abstract do not correlate: what is the cause, what is the consequence or context. Please review.
  3. Still the same missing CSR. If you would check your Discussion, CSR even not mentioned. In the introduction you state clearly that what you are researching is just one of many elements in CSR. That's the impression after the read.  I would still suggest another journal (human capital development, values and culture oriented or the like; the article is good, but for another journal).

Reviewer 2 Report

Please do manuscript formatting according to the journal style.

Author Response

The formatting according to the journal style has been checked and corrected.

Reviewer 4 Report

Dear Authors

Thank you for the extensive changes you have made to your manuscript.
I think it has improved. I recommend reading the discussion section for some language corrections. First you start with the short value scale of Schwartz, and you
continue with 4 value constructs (dimensions). Explain your analysis.
Why not start with a factor analysis and continue with these factors
to investigate the second order constructs and their relationship
to the "rank" to which a person belongs? Note: do you investigate 'cuasality' or a relationship between constructs?
It is not because one has a higher rank; that this 'causes' a new value pattern? Use the right words to claim results and relationships. In the discussion section you come up with values ​​of an entire country and even values ​​of other neighboring countries. Why? You're investigating a military corps ... not the entire population.

some language checks are needed: eg. 

pg. 6: the group

Round 3

Reviewer 1 Report

The same issue of missing CSR as a core element of the article.

Good article, but not for this journal.